# Detection of Pharmaceutical Residues in Surface Waters of the Eastern Cape Province

**DOI:** 10.3390/ijerph17114067

**Published:** 2020-06-07

**Authors:** Sesethu Vumazonke, Sandile Maswazi Khamanga, Nosiphiwe Patience Ngqwala

**Affiliations:** 1Environmental Health and Biotechnology Research Group, Division of Pharmaceutical Chemistry, Faculty of Pharmacy, Rhodes University, P.O. Box 94, Grahamstown 6140, South Africa; g17v8833@ru.ac.za; 2Division of Pharmaceutics, Faculty of Pharmacy, Rhodes University, P.O. Box 94, Grahamstown 6140, South Africa; s.khamanga@ru.ac.za

**Keywords:** pharmaceutical residues, ELISA, lyophilisation and SPE, UPLC-ESI-MS/MS, river water quality

## Abstract

Pharmaceuticals are emerging contaminants in the aquatic environments. Their presence poses toxicological effects in humans and animals even at trace concentrations. This study investigated the presence of antibiotics, anti-epilepsy and anti-inflammatory drugs in river water of selected rivers in the Eastern Cape Province in South Africa. Enzyme-linked immunosorbent assay was used for screening of sulfamethoxazole and fluoroquinolones antibiotics. The samples were collected in upper-stream, middle-stream and lower-stream regions of the rivers and effluent of selected wastewater treatment plants. Pre-concentration of the samples was conducted using lyophilisation and extraction was conducted using solid phase extraction (SPE) on Waters Oasis hydrophilic-lipophilic-balanced cartridge. The percentage recovery after sample clean-up on SPE was 103% ± 6.9%. This was followed by ultra-performance liquid chromatography electrospray ionization tandem mass spectrometry. The detected analytes were sulfamethoxazole, erythromycin, clarithromycin and carbamazepine. Carbamazepine and erythromycin were detected in high concentrations ranging from 81.8 to 36,576.2 ng/L and 11.2 to 11,800 ng/L respectively, while clarithromycin and sulfamethoxazole were detected at moderate concentrations ranging from 4.8 to 3280.4 ng/L and 6.6 to 6968 ng/L, respectively. High concentrations of pharmaceuticals were detected on the lower-stream sites as compared to upper-stream sites.

## 1. Introduction

Pharmaceuticals have been detected in environmental samples such as surface water, groundwater, seawater, sediments and drinking water [1,2,3,4,5,6,7,8,9,10], and they are referred to as emerging contaminants [2,11]. Owing to their broad application in human and veterinary medicine, large amounts of pharmaceuticals are produced yearly [1,11]. Pharmaceuticals such as antibiotics have an estimated consumption of 100,000 to 200,000 tons per year globally [3,12,13,14]. Approximately 5% to 90% of the ingested antibiotic doses are excreted via urine or faeces as a metabolite or parental compound depending on the chemical properties of the compound [3,11,14,15,16,17,18]. These pharmaceuticals end up in sewage systems and eventually enter the environment through sewage leakages, discharge of effluents from wastewater treatment plants (WWTP) which enter the aquatic systems, or through the disposal of unused or unfinished medication [1,19]. The use of sludge and animal manure in agriculture as fertilizer may also lead to contamination of the agricultural soils and may result in the entry of antibiotics into the aquatic systems by leaching into the underground water [3,20]. This may result in contamination of surface water (river, dams and streams) and underground water, which are the main sources of drinking water [21]. This raises concern about the quality of drinking water [22].

The presence of pharmaceutical residues in the environment can be problematic as some of these pharmaceuticals are persistent and can make their way to humans via the food-chain or drinking water [23,24]. The constant exposure of pharmaceuticals to aquatic environments can lead to chronic effects such as alterations in the metabolic or/and reproductive systems in non-targeted organisms [6,25]. Antibiotics in the environment may lead to the development of antibiotic-resistant microorganisms even at low concentration, therefore posing a health concern for both humans and animals since infections caused by antibiotic-resistant bacteria are difficult to treat. Some of the antibiotics persist in the environment, lasting up to months [3,26,27]. Water is essential to life in both plants and animals [28], and thus access to clean and safe drinking water is essential to maintain a healthy life [29] and monitoring of water quality in these water resources is crucial for the protection of public health [6,10,11].

The enzyme-linked immunosorbent assay (ELISA) technique is one of the traditional techniques used to screen for the presence of antibiotic residues in meat, milk, surface water, groundwater, wastewater, soil and manure [30,31,32,33,34,35,36]. ELISA techniques are useful for the screening of structurally similar antibiotic mixtures in a sample [37]. The compounds with similar structures are difficult to differentiate with immunoassays, therefore, liquid chromatography mass spectrometry (LC/MS) or liquid chromatography with tandem mass spectrometry (LC-MS/MS) techniques are used for detection and quantification of structurally similar compounds [27].

Liquid chromatography mass spectrometry has been a method of preference for analysis of pharmaceuticals in environmental samples [2,6,24,38,39]. This is due to the method being suitable for the analysis of polar organic compounds and offers an advantage of rapid analysis of pharmaceuticals in environmental samples [24,40]. LC-MS has high sensitivity [3], selectivity and robustness [6,41]. The high sensitivity of LC-MS/MS with a triple quadrupole (QqQ) analyser is the reason this method is convenient for the detection of pharmaceuticals in surface waters up to parts per billion (ppb) through target analysis in multiple reaction monitoring mode (MRM) [6,10,42,43]. The ionisation method commonly used in LC-MS/MS analysis is electrospray ionisation (ESI) and this is due to its high sensitivity, reliability and robustness. ESI is an ionisation technique widely utilised in production of gas phase ion of thermally labile macromolecules. Analysis using the MRM allows for identification, confirmation, quantification and caters for low detection limits, which may be a result of an increase in the signal-to-noise ratio [2,44].

Low concentrations of pharmaceuticals in environmental samples make a direct analysis of pharmaceuticals by chromatographic techniques challenging, and thus, pre-concentration of the sample is fundamental. The pre-concentration step is not only essential for providing means of detecting pharmaceuticals at low concentrations but also reduces the matrix effect during LC-MS analysis [43]. Solid phase extraction (SPE) is the preferred sample preparation technique due to the small volume of organic solvent required for extraction and cleaning the column, it requires short sample preparation time, it provides pre-concentration of the sample, it is easy to isolate target compounds, it ensures reproducibility and it allows for optimisation and sample clean-up in different sample matrix before chromatographic analysis [2,37,44].

Nevertheless, due to high costs that are involved in LC/MS or LC-MS/MS methods, the technique cannot be used for routine analysis of the pharmaceuticals in environmental samples; therefore, ELISA techniques are usually used [27]. The frequent monitoring of the antibiotics in environmental samples is essential as it provides information about the antibiotics present in the environment, their concentration and the potential of causing negative effects to the environment [37]. The use of ELISA allows for simultaneous screening of various samples within a short period of time and at low costs. The antibodies used in ELISA are usually designed to target one analyte, but they are class-specific and therefore they display a high degree of cross-reactivity with other structurally similar compounds [36,37,45,46,47].

This study investigated the presence of antibiotics, anti-epilepsy and anti-inflammatory drugs in river water of the Eastern Cape Province in South Africa. The drugs investigated were sulfamethoxazole, ciprofloxacin, erythromycin, clarithromycin (antibiotics), carbamazepine (anti-epilepsy) and ibuprofen (anti-inflammatory).

## 2. Materials and Methods

### 2.1. Description of Sampling Sites

Sample collection was conducted in the autumn and spring seasons of 2018 at Buffalo River, Tyhume River, Palmiet River, Bloukrans River and Swartkops River. In each river, four sites were targeted, except for Palmiet River, which was used as a reference site for Bloukrans River. The site names and coordinates are shown in Table 1. The coordinates of the sampling sites were obtained using Google Earth Map.

### 2.2. Sample Collection and Preparation

River water samples were collected using sterilized scotch bottles which were autoclaved at 121 °C for 30 min. Before sample collection, the bottles were rinsed with river water twice and approximately 1 L water sample was collected from each site at a depth of approximately 0.5 m from the water surface in an opposite direction of the water current. Samples were properly labelled, stored in ice and transported into the laboratory for analysis. Prior to sample analysis, water samples were filtered through 0.45 µm pore-sized membrane filters purchased from Merck Millipore (Gauteng, South Africa) and stored at −4 °C. Analysis of the samples was conducted within 48 h. For ELISA screening, the samples were collected twice in each river to ensure concentrations of pharmaceuticals were properly represented.

### 2.3. Preparation of Standards

The target pharmaceutical compounds in this study were carbamazepine, ibuprofen, ciprofloxacin, sulfamethoxazole, erythromycin and clarithromycin, and were purchased from Merck Millipore (Gauteng, South Africa). The target compounds were selected based on global use. The standard solution (1000 µg/mL) of the target compounds were prepared by dissolving the weighed standard powder into methanol (High Performance Liquid Chromatography gradient-grade) purchased from Merck Millipore (Gauteng, South Africa). The structure and molecular weight of the target pharmaceuticals are shown in Table 2.

### 2.4. ELISA Screening

For ELISA screening, the filtered samples were transferred into sterile vials purchased from Merck Millipore (Gauteng, South Africa) and kept on ice. The screening was performed using ELISA kits purchased from Abraxis LLC (Warminster, England). Screening of fluoroquinolones was performed using the screening kit following the manufacturer’s protocol. Approximately 50 µL of the standard solutions, samples, enzyme conjugate and antibody solution were added to the ELISA microtiter plate pre-coated with secondary antibodies (goat anti-rabbit) specific to a unique antigenic site on the fluoroquinolone molecule. The solution was mixed properly, and the plate was incubated for 1 h at room temperature. Following incubation, the wells were washed 3 times with 1X wash buffer solution and tapped dry on a stack of paper towel. Approximately 100 μL of the substrate (colour) solution was added to the wells and incubated at room temperature for 20–30 min, 100 μL of the stop solution was added to the wells to stop the enzyme reaction. The screening for sulfamethoxazole was performed using the sulfamethoxazole ELISA kit following the manufacturer’s protocol. Approximately 75 µL of the samples, control and standards were added to the microtiter plate pre-coated with goat anti-rabbit. About 50 μL of the anti-sulfamethoxazole antibody solution was added to each well. The solution was then mixed for 20–30 s and the plate was incubated at room temperature for 20 min. About 50 μL of the sulfamethoxazole enzyme conjugate solution was added to each well. The solution was mixed properly for 20–30 s and the plate was incubated for 40 min at room temperature. The wells were washed 3 times with 1× wash buffer solution and tapped dry on a stack of paper towels. After wash, 150 μL of the substrate (colour) solution was added and the plate was incubated for 30 min at room temperature. About 100 µL of the stop solution was added after incubation to stop the enzyme reaction. Both fluoroquinolones and sulfamethoxazole optical density was read at 450 nm with a microtiter plate reader. The zero standard (0 ppb) results in the maximum binding of the enzyme conjugate. The results were reported in a percentage of zero standards. Interpretation of the results was performed manually by plotting a standard curve using the results obtained for the standards, and the concentrations of the samples were read from this curve.

### 2.5. Concentration of Samples by Lyophilization and SPE Extraction

Prior to chromatographic analysis, the pre-concentration of water samples was conducted by the freeze-drying method. About 500 mL of the filtered river water samples were frozen using liquid nitrogen on a round bottom flask connected to a Buchi Rotavapor, model R-215 (Labortechnik AG, Flawil, Switzerland), at a speed of 160 rpm. The samples were frozen until they reached a temperature of approximately −40 °C and dried on a freeze-dry machine VirTis BenchTop K, model #2KBTES-55 (SP Scientific, Warminster, PA, United States). The freeze-dryer condenser temperature was set at −50 °C and the vacuum was set at 300 mT. The samples were left on a freeze-dryer machine until a powder sample was obtained. The product yield was weighed and transferred into a sterile 10 mL vial and labelled properly. The lyophilised samples were sent to the University of Stellenbosch for ultra-performance liquid chromatography coupled with electron spray ionizer tandem mass spectrometry (UPLC-ESI-MS/MS) analysis (Waters Corporation, Winslow, UK). Prior to sample analysis, the lyophilised samples were reconstituted in 9 mL of 10% methanol consisting of 1 mL of 50 g/L p-aminosalicylic acid used as an internal standard. Pharmaceutical residues were isolated from the sample by passing the 10 mL sample through SPE cartridge, packed with an Oasis hydrophilic-lipophilic-balanced (HLB) 6 cc Vac cartridge, 200 mg sorbent, purchased from Waters (Milford, MA, USA). The cartridge was washed with water and the analytes were eluted off using 1 mL methanol. The percentage recovery of the analytes after sample clean-up was 103% ± 6.9%.

### 2.6. Chromatographic Separation of Pharmaceuticals

Chromatographic separation was carried out on an Acquity UPLC linked to a Xevo TQS triple quadrupole Mass Spectrometer (Waters, Corporation, Winslow, United Kingdom). The analytes were separated on 2.1 × 100 mm, 1.7 µm UPLC ethylene bridged hybrid (BEH) reverse phase C18 analytical column (Waters, Johannesburg, South Africa). The mobile phase consisted of solvent A: 0.1% formic acid in water and solvent B: 0.1% formic acid in acetonitrile. The column was initially eluted using 0.1% formic acid in water for 0.5 min at a flow rate of 0.3 mL/min, then increased to 0.1% formic acid in acetonitrile for 9.5 min at a flow rate of 0.3 mL/min then back to 0.1% formic acid in water for 1.5 min at a flow rate of 0.3 min. Before the next injection, the column was allowed to calibrate for 5 min. The analysis was carried out for 12 min and the retention times were between 4.59 to 5.70 min for the four detected analytes. Mass spectrometry was performed using a Xevo TQS triple quadrupole Mass Spectrometer (Waters Corporation, Winslow, UK) equipped with an electrospray ionisation source. The mass spectrometer was carried out in MRM, and the cone voltage and collision energy were optimised for each analyte in positive ionisation mode. The instrumental parameters for the UPLC-ESI-MS/MS method are shown in Table 3.

## 3. Results

### 3.1. ELISA Screening

In this study, the presence of pharmaceutical residues in river water of the Eastern Cape Province was investigated. The concentration of sulfamethoxazole and fluoroquinolone screened by ELISA are presented in Table 4. Sulfamethoxazole (SMX) was detected in 12 out of 13 samples with 92.3% detection frequency, while fluoroquinolones were detected in 8 out of 13 samples with 61.54% detection frequency. The detection frequency for SMX observed in this study was slightly higher compared to the previous study, where 88.9% detection frequency was observed [27]. SMX was obtained in high concentrations ranging from 0 to 1400 ng/L, while fluoroquinolones were obtained in a concentration ranging from 0 to 500 ng/L. The concentrations of sulfamethoxazole observed in this study in river water samples (Table 4) was much higher compared to the concentrations which were reported in a previous study, ranging from 10 to 90 ng/L [27]. However, the opposite was observed in WWTP samples in North Dakota in the United States, where higher concentrations of SMX (1100 to 2500 ng/L) were reported by Shelver et al. [27] compared to this study, while smaller concentrations (76 to 170 ng/L) were reported by Hendrick and Pool. [13]

### 3.2. Pharmaceutical Detection by UPLC-ESI-MS/MS

The pharmaceuticals detected by UPLC-ESI-MS/MS from the analysed river samples were carbamazepine, erythromycin, sulfamethoxazole and clarithromycin. Ibuprofen and ciprofloxacin were the two target pharmaceuticals that were not detected by UPLC-ESI-MS/MS in all river water samples and the results are presented in Table 5. Carbamazepine was the only pharmaceutical present in all the samples, with 100% detection frequency. Similar results have been reported in the Llobregat River in Spain, where the concentration range of 8 to 179 ng/L was observed [6]. Clarithromycin was present in all the samples, except for site P1 at Palmiet River, and had 93.75% detection frequency. Erythromycin and sulfamethoxazole were detected in 11 and 12 of the 16 analysed samples, with a detection frequency of 68.75% and 75%, respectively. As shown in Table 5, erythromycin was the only pharmaceutical detected in high concentrations, ranging from not detected (P1, B1, B2, S1 and T1) to 744.2 ng/L (Bl3). Carbamazepine followed with a concentration range of 81.8 (P1) to 36,576.2 ng/L (S3).

## 4. Discussion

### 4.1. ELISA Screening

SMX and fluoroquinolones concentrations observed were lower than the detection limit (0.015 µg/L) of the ELISA test kits used at site B1 at Buffalo River. While for fluoroquinolones, concentrations lower than the detection limit were observed at Buffalo River (B1 and B2), Palmiet River (P1) and Swartkops River (S1 and S2), as shown in Table 4. The results obtained for SMX correspond to the results which were previously reported by Shelver et al. [27], where concentrations lower than detectable limits were reported for river water samples. The SMX concentration observed at site S1 in Swartkops River can be explained by the fact that this site is situated in the Groendal nature reserve. SMX is one of the commonly used antibiotics in human and animal medicine. It is possible that people who visit this nature reserve might have urinated near the river or improperly disposed of this drug, or it is probably used in wild animals and through surface runoff it ended up in river water [49,50]. SMX is resilient to degradation, and thus it lasts in the environment and has been reported in aquatic systems by several authors [13,51,52,53]. High concentrations of fluoroquinolones were observed at middle-stream and lower-stream site samples, except at T1 site in Tyhume River. This was an indication that human contamination and WWTP are the main sources of fluoroquinolones in surface water. The presence of fluoroquinolones has been reported in WWTP influent and effluent samples at a concentration range of 300 to 500 ng/L in France, Greece and Italy, and 30 to 1100 ng/L in Switzerland, while in South Africa, concentrations of 89 to 92 ng/L were reported [13]. The concentration of fluoroquinolones observed in the WWTP samples was similar to the ones observed in France. The release of fluoroquinolones into aquatic bodies can negatively impact fish species as well as consumers of fish [13,49,50].

### 4.2. Pharmaceuticals Detected by UPLC-ESI-MS/MS

Ibuprofen is a commonly used anti-inflammatory drug, therefore it was expected to be present in river water at high concentrations, more especially in WWTP effluent samples (Table 5). However, it was not detected, and this may be due to its poor fragmentation, since confirmation of pharmaceuticals on UPLC-ESI-MS/MS (QqQ) requires two selected reaction monitoring transitions. Ibuprofen can only produce one product ion from the precursor ion, thus making it undetectable by QqQ mass spectrometry. Similar results were previously reported by López-Roldán et al. [6] in river water samples where ultra-performance liquid chromatography–mass spectrometry with a time-of-flight analyser and liquid chromatography-tandem mass spectrometry with a triple quadrupole analyser were used to analyse river water samples. Ciprofloxacin is one of the commonly used antibiotics to treat various bacterial infections and its wide use in aquaculture is the reason it is found in environmental samples [2,13,54]. Ciprofloxacin was not detected in all the samples and this was most likely due to the UPLC-ESI-MS/MS library, which did not have ciprofloxacin, thus causing it to fall outside the library spectrum. This indicates that ciprofloxacin was probably present in the samples but not detected. Also, ciprofloxacin falls within the class of fluoroquinolones antibiotics and was positively screened with ELISA.

Carbamazepine is extremely persistent in the environment, with the removal rate up to 10% by the sewage treatment plants (STPs) [55]. The presence of carbamazepine in all river water samples was an indication that carbamazepine is one of the commonly used drugs in the Eastern Cape Province. Higher concentrations of carbamazepine were obtained from the effluent samples of WWTPs compared to other sites in all the studied rivers. This may be due to WWTPs being inefficient in removing carbamazepine from wastewater, and it is therefore discharged with wastewater into surface waters [55,56]. This indicates that the municipal waste and hospital waste are the main sources of carbamazepine in WWTPs. Higher concentrations of carbamazepine were obtained from the Swartkops River samples as compared to other rivers, indicating that carbamazepine is one of the commonly used drugs at Uitenhage, with concentration ranging from 2261.4 (S1) to 36,576.2 ng/L (S3). The lowest concentrations of carbamazepine were obtained from the upper-stream sites in all the rivers, except for the Swartkops River upper-stream site. No health effects have been reported for the concentration of carbamazepine obtained at P1, B1 and T1 sites. A high concentration of carbamazepine observed at site S1 may be explained by the site being inside the Groendal nature reserve and that the people who visit this site may be using carbamazepine. The presence of carbamazepine in river water may be as a result of excretion via urine or improper disposal. Carbamazepine has been reported to affect the circulating thyroid hormones at a concentration of 179 ng/L [6] and the ultrastructure of the fish kidney, liver and gills at a concentration of 1000 ng/L [56,57]. Concentrations that were obtained, except at site P1 and T1, were higher than 179 ng/L, and according to López-Roldán et al. [6], such concentrations can lead to a disturbance in circulating thyroid hormones. In most of the sites, the concentration of carbamazepine was above 1000 ng/L, and these concentrations affect the ultrastructure of the fish kidney, liver and gills [55,58]. The presence of carbamazepine in surface waters has been reported worldwide.

SMX was detected with a concentration range of not detected (B2, S1 and T1) to 6968 ng/L (Bl3). However, at B2, S1, T1 and T4 sites, SMX was not detected since the concentrations were lower than the detection limits, although it was detected by ELISA screening at these sites. High concentrations of SMX observed at middle stream and WWTP samples was an indication that they are the sources of SMX in surface water. The presence of SMX in the environment has been reported to change the thyroid function at a concentration equal to or higher than 119 ng/L [6]. In all the sites, the SMX concentration was higher than 119 ng/L, indicating that the river water had the potential of affecting human and animal health following exposure [6].

In all the antibiotics that were detected in river water samples, erythromycin was the only antibiotic detected in higher concentrations. A high concentration of erythromycin was obtained in effluent samples of WWTPs in all the rivers, except for site S2 in the Swartkops River. The absence of erythromycin on upper-stream sites of the rivers and B2 was an indication that these sites were not contaminated by erythromycin even though these sites were contaminated by clarithromycin. High concentrations of erythromycin were obtained in effluent samples of WWTPs in all the rivers except for site S2 in Swartkops River. A high concentration at S2 can be explained by effluent discharge from the WWTP located upstream and that this site is near the houses, so improper disposal, sewage leakage and surface runoff might have contributed to the high concentrations obtained. The high concentrations of erythromycin observed in WWTP effluent samples were an indication that the WWTP processes are inefficient in removing erythromycin and that WWTPs are the main source of erythromycin in surface waters [2,38,59]. The high concentrations observed in middle-stream sites and WWTPs sites were an indication that erythromycin is one of the commonly used antibiotics since it is used to treat a variety of bacterial infections. Lower concentrations observed in lower-stream sites were due to dilution by river water as the river is flowing further down. However, at Swartkops River, a high concentration of erythromycin was observed at the lower-stream site, and this may be due to the accumulation of the small quantities of erythromycin at this site because of the decrease in velocity of the water. The concentration of erythromycin obtained in this study was higher than the concentrations which were reported in Sweden influent and effluent samples of WWTP [60].

Clarithromycin was detected in all the samples, and this was an indication that clarithromycin is one of the commonly used antibiotics since it is used to treat a variety of bacterial infections (Table 5). Clarithromycin was observed at a concentration range of 4.8 (B1) to 3280.4 ng/L (Bl2). Higher concentrations of clarithromycin were detected in effluent samples of WWTPs at Buffalo River and Tyhume River, while at Swartkops River and Bloukrans River, higher concentrations were obtained on middle-stream sites (S2 and Bl2). The high concentration at this site at Bloukrans River may be due to sewage leakage, while at Swartkops River, it may be due to effluent discharge from the WWTP upstream. A low concentration of clarithromycin at lower-stream sites of the rivers was due to dilution by river water as the river flows further down. Clarithromycin concentration observed in this study was less than 2 µg/L in all the sites, indicating that the risk of toxicity to eukaryotic algae [60,61] and other aquatic organisms was minimal. However, clarithromycin can have negative effects on the microbial community and lead to antibiotic resistance, even at these concentrations.

## 5. Conclusions

This study reported a successful screening of the presence of fluoroquinolones and sulfamethoxazole in river water. The detection of pharmaceuticals in river water of the Eastern Cape Province was an indication that these rivers are polluted by pharmaceuticals. High concentrations in the middle-stream, WWTP effluents and lower-stream sites was an indication that human activities are the main source of pollution in those rivers. Therefore, constant monitoring of the concentration of pharmaceutical contaminants is essential to protect public health, aquatic biodiversity and the quality of river water.

## Figures and Tables

**Table 1 ijerph-17-04067-t001:** Co-ordinates of the sampling sites of all the studied rivers.

River	Site No.	Site Name	Latitude	Longitude
Palmiet	P1	Upper-stream	33.369625	26.476542
	Bl2	Middle-stream	33.314295	26.551907
Bloukrans	Bl3	Wastewater treatment plant	33.316896	26.559300
	Bl4	Lower-stream	33.317766	26.568247
	B1	Upper-stream	32.789741	27.369707
Buffalo	B2	Middle-stream	32.896940	27.392820
	B3	Wastewater treatment plant	32.900088	27.404174
	B4	Lower-stream	32.934406	27.440321
	T1	Upper-stream	32.610883	26.909413
Tyhume	T2	Middle-stream	32.797323	26.847497
	T3	Wastewater treatment plant	32.792744	26.849658
	T4	Lower-stream	32.827368	26.888672
	S1	Upper-stream	33.716609	25.288034
Swartkops	S2	Middle-stream	33.791756	25.407598
	S3	Wastewater treatment plant	33.784194	25.426816
	S4	Lower-stream	33.792082	25.490763

**Table 2 ijerph-17-04067-t002:** List of target pharmaceutical compounds and their structures [48].

Compound Group	Target Compound	Use	Molecular Weight (g/mol)	Structure
Anticonvulsants	Carbamazepine	Anti-epileptic	236.274	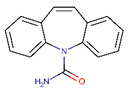
Non-steroidal anti-inflammatory drug	Ibuprofen	Anti-inflammatory	206.285	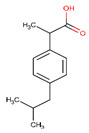
Macrolide antibiotic	Erythromycin	Antibiotic	733.937	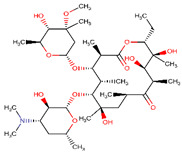
Macrolide antibiotic	Clarithromycin	Antibiotic	747.953	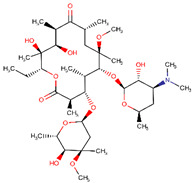
Sulfonamide	Sulfamethoxazole	Antibiotic	253.276	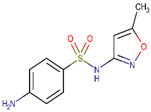
Fluoroquinolone	Ciprofloxacin	Antibiotic	331.347	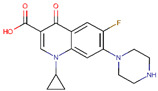

**Table 3 ijerph-17-04067-t003:** The analytical parameters for the ultra-performance liquid chromatography coupled with electron spray ionizer tandem mass spectrometry (UPLC-ESI-MS/MS) method.

Pharmaceutical	Limit of Detection (ng/L)	Limit of Quantification (ng/L)	Parent Compound (m/z)	Daughter Compound (m/z)	Cone Voltage	Collision Voltage	Ion Mode
Carbamazepine	0.1	0.1	273.0000	135.0000	20	10	ESI +
Ibuprofen	0.5	1.4	ND	ND	ND	ND	
Erythromycin	0.9	2.3	734.0000	158.0000	15	15	ESI +
Ciprofloxacin	1.0	3.4	ND	ND	ND	ND	
Sulfamethoxazole	0.3	0.9	254.0000	147.0000	20	25	ESI +
Clarithromycin	<0.1	<0.1	748.8000	590.6000	30	20	ESI +

ND: not detected.

**Table 4 ijerph-17-04067-t004:** Concentration of sulfamethoxazole and fluoroquinolones of Buffalo, Bloukrans, Swartkops and Tyhume River water samples detected by enzyme-linked immunosorbent assay (ELISA) screening during the autumn season.

River	Season	Site Number	Sulfamethoxazole (ng/L)			Fluoroquinolones (ng/L)		
			Minimum	Average	Maximum	Minimum	Average	Maximum
**Buffalo**	**Autumn**	B1	ND	ND	ND	ND	ND	ND
		B2	640	700	760	ND	ND	ND
		B3	NS	NS	NS	NS	NS	NS
		B4	880	900	920	79	100	121
**Palmiet**	**Autumn**	P1	178	200	222	ND	ND	ND
**Bloukrans**	**Autumn**	Bl2	1254	1400	1546	319	400	481
		Bl3	1300	1400	1500	497	500	503
		Bl4	1281	1300	1319	123	400	277
**Swartkops**	**Autumn**	S1	85	100	115	ND	ND	ND
		S2	987	1100	1213	ND	ND	ND
		S3	NS	NS	NS	NS	NS	NS
		S4	1096	1200	1304	387	400	413
**Tyhume**	**Autumn**	T1	90	100	110	266	300	334
		T2	180	200	220	219	200	181
		T3	NS	NS	NS	NS	NS	NS
		T4	430	400	370	235	200	165

NS: not sampled; ND: not detected.

**Table 5 ijerph-17-04067-t005:** The pharmaceutical residues detected with UPLC-ESI-MS/MS in river water samples collected from Buffalo, Bloukrans, Swartkops and Tyhume Rivers during the spring season.

		Class					
		Anti-Epilepsy	Macrolide Antibiotic	Antibiotic	Macrolide Antibiotic	Antibiotic	Anti-Inflammatory
River	Site no.	Carbamazepine (ng/L)	Erythromycin (ng/L)	Sulfamethoxazole (ng/L)	Clarithromycin (ng/L)	Ciprofloxacin (ng/L)	Ibuprofen (ng/L)
**Palmiet**	P1	81.8	ND	6.6	ND	ND	ND
**Bloukrans**	Bl2	14,363.6	533.6	4535.2	3280.4	ND	ND
	Bl3	26,329.6	744.2	6968	1541.8	ND	ND
	Bl4	9717.8	163.6	5974.4	478.2	ND	ND
**Buffalo**	B1	216	ND	20.2	4.8	ND	ND
	B2	337.8	ND	ND	8.4	ND	ND
	B3	14,963.8	263	3645.4	315	ND	ND
	B4	4758	83.6	1305.4	67.4	ND	ND
**Swartkops**	S1	2261.4	ND	ND	144	ND	ND
	S2	5291.2	11,800	2918.8	264.8	ND	ND
	S3	36,576.2	34.6	1979.8	260.8	ND	ND
	S4	17,345.2	60.6	2666.4	98.2	ND	ND
**Tyhume**	T1	123.4	ND	ND	5.4	ND	ND
	T2	943.8	28.4	334.4	82	ND	ND
	T3	5097.6	117.8	913.2	440	ND	ND
	T4	2300.2	11.2	ND	13.8	ND	ND

ND: not detected.

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
