# Peer review of "Detection of Pharmaceutical Residues in Surface Waters of the Eastern Cape Province"

_ijerph, 2020, doi:10.3390/ijerph17114067_

Round 1

Reviewer 1 Report

Detection of pharmaceutical residues in surface waters of the Eastern Cape Province was conducted, and it is meaningful, the research finally pointed out the importance of constant monitoring of the concentration of pharmaceutical contaminants in these rivers to protect public health and aquatic biodiversity. But the paper needs some modifications before accepting.   1. If possible, the monitoring data can be more detailed. The monitoring data of all seasons can be used. If not, please explain why autumn and spring were selected and their representativeness.   2. In the data statistics, the average value was simply used. It is suggested to supplement the minimum value and the maximum value, and cooperate with the average value at the same time, so as to better and more comprehensively elaborate the pollutant concentration level in these rivers.   3. It is suggested to supplement and refine the sampling section and explain what measures were taken to ensure the representativeness of the samples.

Author Response

Please see the attachment regarding the comments. The responses are highlighted .

Reviewer 2 Report

The present manuscript focuses on the pharmaceutical detection in surface waters. Indeed, the investigated topic represents a concernig issue from both an environmental and health perspective. However, the work needs to be significantly improved in order to meet suitable requirements for potential publication. Please find as follows some recommendation: 

Comment #1 (Page 1): "...surface water, groundwater, seawater, sediments and drinking water [1-10]...", here are ten grouped references for a single sentence, however the present manuscript is not a review paper. Please be not recurring with the literature references trying to report the required and most current ones.   

Comment #2 (Page 2): "Traces of pharmaceutical residues have been reported in drinking water by several authors [1, 2, 6, 10, 11, 22, 23] thus raising concern on the quality of drinking water [24].", this is already stated in the first part of the Introdcution. There should not be redundancy in the text.

Comment #3 (Introduction): This section should be revised in a more concise version. Some concept is quite repeated while repetitions should be avoided. Instead, this section does not highlight the novel aspect of this work. Authors should focus on the statement of their work novelty and highlight the different perspectives of their work compared to previous ones in this research field.

Comment #4 (Materials and methods section): please provide a proper partition of this Section in different Sub-sections (e.g. 2.1, 2.2, etc.). This section requires to be better structured.

Comment #5 (Page 5): "The optical density was read at 450 nm with a microtiter plate reader. The zero standard (0 ppb) results in the maximum binding of the enzyme conjugate. The results were reported in a percentage of zero standards. Interpretation of the results was performed manually by plotting a standard curve using the results obtained for the standards and the concentrations of the samples were read from this curve.", this is repeated twice. If the procedure is the same, it can be referred to both analysis when reported by avoiding copied sentences in the manuscript.

Comment #6 (Page 6): "2.1. ELISA screening", there is a mistake in this sub-section numbering. 

Comment #7 (Conclusions section): this section mainly reports information highlighted in the discussion section. Authors should revise it in order to avoid repetitions.

Author Response

Please see the attachment regarding the comments. The response are highlighted .

Round 2

Reviewer 1 Report

this paper can be accepted now.

Reviewer 2 Report

The Authors properly addressed to the suggested revisions. The manuscript can be considered as acceptable for publication in its current form.